# Animal invaders threaten protected areas worldwide

Xuan Liu [1✉], Tim M. Blackburn [2,3], Tianjian Song[1,4], Xuyu Wang[5], Cong Huang[6] & Yiming Li [1,4✉]

Protected areas are the cornerstone of biodiversity conservation. However, alien species invasion is an increasing threat to biodiversity, and the extent to which protected areas worldwide are resistant to incursions of alien species remains poorly understood. Here, we investigate establishment by 894 terrestrial alien animals from 11 taxonomic groups including vertebrates and invertebrates across 199,957 protected areas at the global scale. We find that <10% of protected areas are home to any of the alien animals, but there is at least one established population within 10-100 km of the boundaries of 89%-99% of protected areas, while >95% of protected areas are environmentally suitable for establishment. Higher alien richness is observed in IUCN category-II national parks supposedly with stricter protection, and in larger protected areas with higher human footprint and more recent designation. Our results demonstrate that protected areas provide important protection from biological invasions, but invasions may become an increasingly dominant problem in the near future.

[1] Key Laboratory of Animal Ecology and Conservation Biology, Institute of Zoology, Chinese Academy of Sciences, 1 Beichen West Road, Chaoyang, 100101 Beijing, China. [2] Centre for Biodiversity and Environment Research, Department of Genetics, Evolution & Environment, University Colledge London, Gower Street, London WC1E 6BT, UK. [3] Institute of Zoology, Zoological Society of London, Regent's Park, London NW1 4RY, UK. [4] University of Chinese Academy of Sciences, 100049 Beijing, China. [5] Institute of Physical Science and Information Technology, Anhui University, Hefei 230601, China. [6] School of Life Science, South China Normal University, Guangzhou 510631, China. ✉email: liuxuan@ioz.ac.cn; liym@ioz.ac.cn

Globally, terrestrial protected areas (PAs) cover about 15% of the Earth's land surface and make priceless contributions to conserving endemic species and maintaining natural ecological functions[1–3]. However, in the current Anthropocene era, the environmental challenges associated with human activities in PAs have attracted considerable public and academic concerns[4,5]. Alien species constitute one such environmental problem and have been identified as a serious threat to global terrestrial PAs[6,7]. Biological invasions by alien species are regarded as one of the top five direct drivers (together with habitat destruction, over-exploitation, climate change and pollution) of recent global biodiversity loss, according to the Intergovernmental Science-Policy Platform on Biodiversity and Ecosystem Services[8]. The impacts of alien species are linked to the declining conservation status of around one-quarter of threatened species[9] and are the leading cause associated with global extinctions since 1500CE[10]. Alien species can also drive the degradation of ecosystem functions by altering trophic interactions, nutrient cycling and habitat structures[11].

Economic and trade globalization cause alien species to continue to establish at ever increasing rates[12] due to rising human movement between regions, and may in consequence also be transported more into PAs throughout the world[13]. For instance, a recent study found that around one third of PAs are under severe pressure from human activities, with the pressure greater on areas with lower levels of protection[5]. Therefore, one might expect there to be more alien species established in less strictly protected PAs due to human activities. Indeed, a continental study based on European Natura 2000 protected areas found that there are more invasive species in PAs with less strict protection[14], but this hypothesis has not been tested at the global scale. The International Union for Conservation of Nature (IUCN) categorizes PAs into strict nature reserves (Ia), wilderness areas (Ib), national parks (II), natural monument or feature (III), habitat/species management area (IV), protected landscape/seascape (V), protected area with sustainable use of natural resources (VI), and those not assigned from the strictest protection to permitting certain human activities[15]. More established alien species would be hypothesized to be found in PAs with lower levels of protection, but this has not been tested across the entire range of IUCN categories of PAs worldwide.

Concerns about the potential threats of alien species in PAs started about 150 years ago, but have been growing rapidly since the 1980s, especially in recent years[7,16]. This has led to global biodiversity conservation fora such as the Convention on Biological Diversity (CBD) and the IUCN World Commission on Protected Areas to call for urgent efforts to prevent alien species incursions into PAs[16]. For instance, although IUCN classifies global PAs in different categories for various management objectives, a primary management goal for all PAs is to reduce invasion risks of alien species and perpetuate the natural state to the greatest possible extent[15]. There is an international SCOPE (Scientific Committee on Problems of the Environment) programme on biological invasions that focuses on a total of 24 PAs across different biomes as cases to monitor the long-term dynamics of alien species invasions in PAs[13]. A recent report showed that most of these PAs carried out short- or long-term plans for managing alien species, but their effectiveness varied across regions and taxa[17]. Previous studies involving taxon-specific, regional or continental analyses have provided valuable insights into the establishment, impact and risk of alien species in PAs[14,16,18]. However, globalization means that the introduction, establishment and spread of alien species generally occur over large spatial scales[19]. A worldwide assessment of the occurrence and potential drivers of alien species in PAs across broad taxonomic groups is therefore critically important for effective mitigation actions, but unfortunately is lacking.

Many factors, including the areas and designation dates for PAs, human activities, native biodiversity, conservation categories and biomes, may affect the number of established alien species in PAs. PAs designated earlier are likely to acquire fewer established alien species owing to their earlier implementation of prevention measures[14,17]. Intense human activities may result in more established alien species due to high propagule pressures driven by the movement of organisms and invasion vectors[20], or due to more vacant niches for alien species establishment created by human activities[21]. PAs with higher native species richness may have fewer aliens due to biotic resistance[22,23]. Large differences in establishment success for alien species exist among taxa and between different regions[19,24]. Yet, we know little about differences in established alien species richness among taxonomic groups in PAs, or about the contributions of these factors to the number of alien species invading PAs at the global scale.

Here, we compile a global database comprising spatially-explicit distributional records of 894 established terrestrial alien animal species from four vertebrate (98 amphibian, 178 reptile, 391 bird and 150 mammal species) and seven invertebrate taxa (66 insects and 11 species from other groups) (Supplementary Data 1 and 2, Supplementary Table 1, Supplementary Fig. 1) . We use this database to quantify the establishment of alien animals in 199,957 terrestrial PAs, and nearby areas, worldwide. We identify the abiotic, biotic and anthropogenic correlates of these incursions by linking alien animal richness in PAs to the designation date and area of PAs, and to human influences and native biodiversity in PAs, and demonstrate the high level of impending risk to uninvaded PAs. Finally, we project the potential richness of the 894 alien species in PAs worldwide based on niches of native and invaded ranges across individual species. The potential richness of PAs is likely to reflect future invasion risks, or environmental conditions suitable for establishment by numerous alien species[25,26]. Our findings suggest that PAs face high potential invasion risks from alien animal invaders due to higher environmental suitability for establishment, and accelerating anthropogenic pressures.

## Results

**Established alien animals and their risks to PAs.** We find that more than 58% (520/894) of alien animal species (mean ± s.e.m. = 0.27 ± 0.003) have become established in PAs, but in only 9.1% (18,110) of the 199,957 PAs (Fig. 1a), even though more than 95% of all PAs are environmentally suitable for establishment by at least some of these species (Fig. 1b). The predicted richness of these alien animal species in PAs (53.34 ± 0.070) on the basis of habitat suitability is 194 times (95% confidence interval: 189–199) higher than the observed richness (two-tailed Wilcoxon signed-rank test, $P < 0.0001$, Fig. 1), suggesting that PAs have far fewer alien animal species incursions than we would otherwise expect, but may be very sensitive to alien animal incursions. Moreover, our analyses reveal that most PAs are at risk of further incursions by aliens: 89.4% of PAs have populations of alien animals established within 10 km of their boundaries (58% of the alien species), while 99.0% of PAs have populations established within 100 km (79.6% of the aliens) (Fig. 2).

**Taxonomic variation in alien animal richness in PAs.** The largest proportion of invaded PAs was colonized by alien birds (4.7% PAs, 252 species), followed by mammals (3.7% PAs, 91 species), invertebrates (2.2% PAs, 63 species), amphibians (0.5% PAs, 48 species) and reptiles (0.4% PAs, 66 species) (Supplementary Fig. 2). The alien species most commonly found

**a** Established richness of alien animals in global protected areas

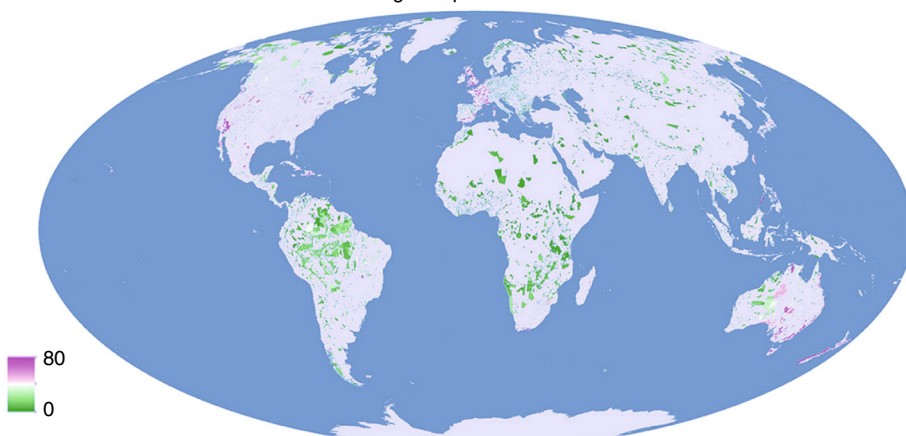

**b** Predicted richness of alien animals in global protected areas

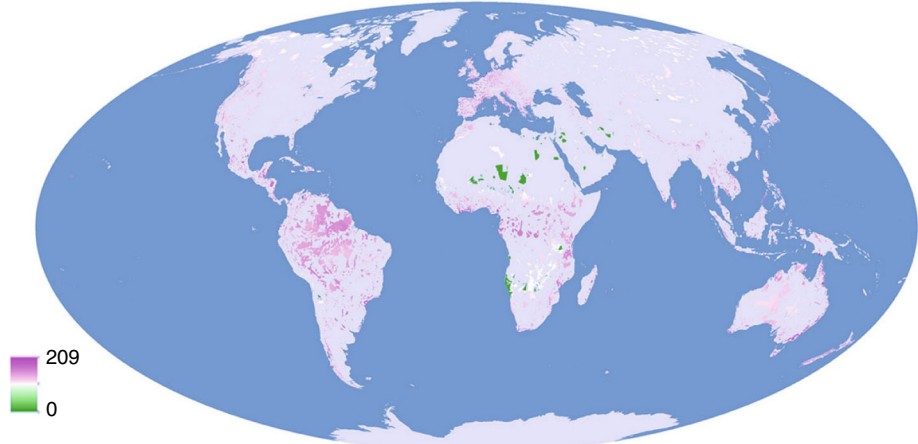

**Fig. 1 Established and predicted richness of 894 alien animals in 199,957 global terrestrial Protected Areas.** Richness was calculated as the total number of species across 11 taxa (Amphibia, Reptilia, Aves and Mammalia, Arachnida, Branchiopoda, Chilopoda, Diplopoda, Malacostraca, Gastropoda and Insecta) established in each protected area. The predicted richness of alien species was determined using species distribution models (SDMs) based on an ensemble of five widely used powerful algorithms. SDMs fit correlative models to alien animal distributions and environmental niches from native and invaded ranges, and then identify the most suitable habitats in PAs.

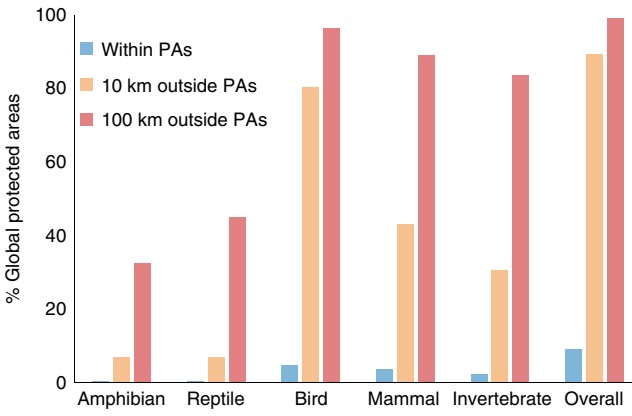

**Fig. 2 Proportion of global terrestrial PAs and the surrounding areas (10 and 100 km distance to PA boundaries) colonized by different taxonomic groups.** We designated PAs as invaded when at least one animal species has established an alien population therein, and also quantified the proportion of uninvaded PAs with at least one animal species with an established population within 10 and 100 km of the PA boundary.

in PAs are all birds: the Rock Dove (*Columba livia*; 6,450 PAs), Ring-necked Pheasant (*Phasianus colchicus*; 4822 PAs) and House Sparrow (*Passer domesticus*; 3972 PAs). Alien species from other taxa commonly found in PAs include, from mammals: Wild Rabbit (*Oryctolagus cuniculus*; 1673 PAs), American Mink (*Neovison vison*; 1251 PAs) and House Mouse (*Mus musculus*; 1,177 PAs); from reptiles: Mediterranean House Gecko (*Hemidactylus turcicus*; 198 PAs) and Red-eared Slider (*Trachemys scripta*; 164 PAs); from amphibians: Cane Toad (*Rhinella marina*; 265 PAs) and American Bullfrog (*Rana catesbeiana* = *Lithobates catesbeianus*; 161 PAs); and from invertebrates: Harlequin Ladybird (*Harmonia axyridis*; 2686 PAs), Africanized Honeybee (*Apis mellifera scutellata*; 527 PAs), and the Yellow Fever Mosquito (*Aedes aegypti*; 284 PAs).

**Spatial variation in alien animal richness in PAs.** The 894 alien animal species in our analysis have established in PAs across various ecosystems worldwide, although they do not distribute homogeneously among continents or biomes (Kruskal–Wallis test, all *P* < 0.0001, Supplementary Table 2). The (sub)tropical Pacific and Caribbean Islands and New Zealand, flooded grasslands and savannas in Florida, and temperate mixed forests, grasslands and savannas in western Europe, Australasia and US

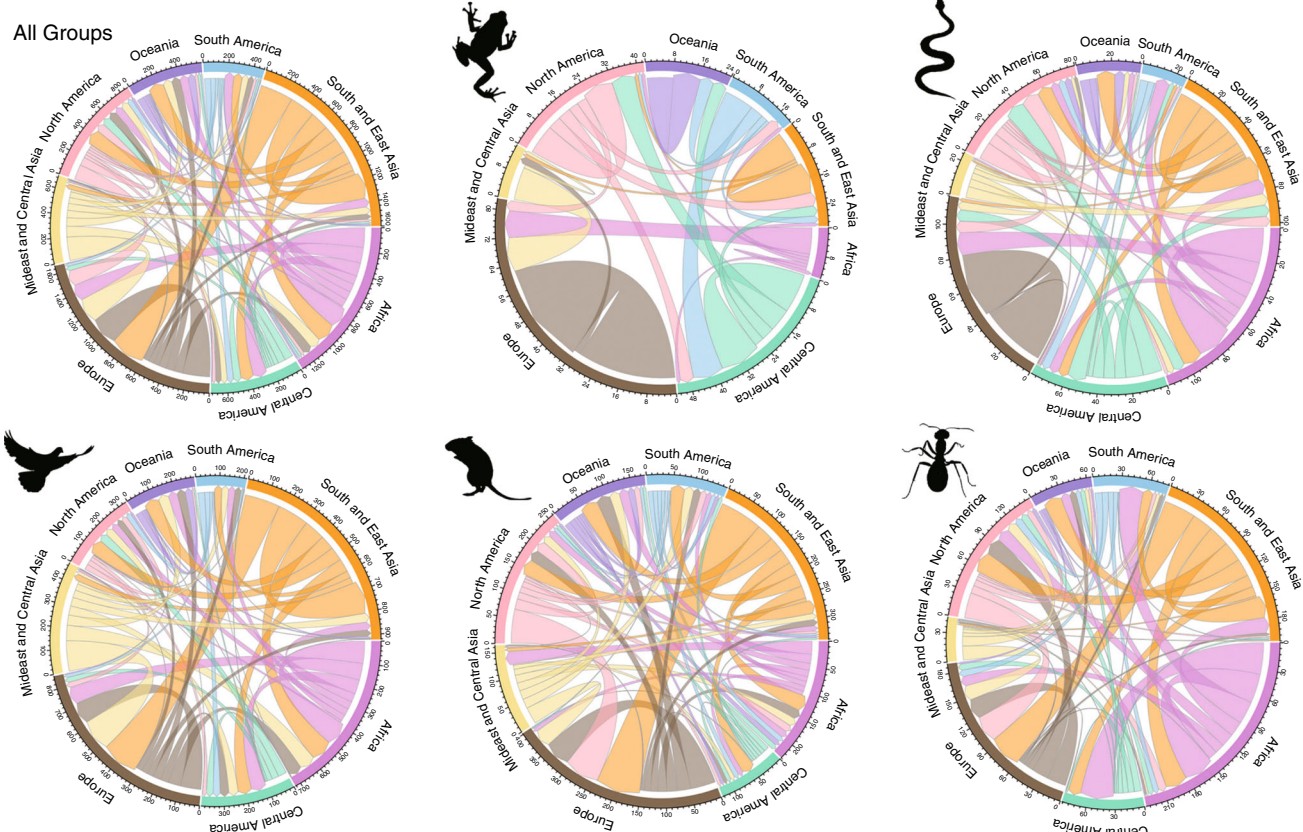

**Fig. 3 Chord diagram showing the global flows of established alien animals in PAs from native to alien continents across taxa.** Colours indicate different continents where the species are native. The width of a chord represents the number of establishments. The size of the outer circle segments indicates the total number of establishments in or originating from the continent. Credit: Tianjian Song (animal silhouettes).

western states, represent higher-richness regions (Fig. 1a), albeit that there are variations in spatial distributions among taxa (Supplementary Fig. 2). Hotspots of alien animal incursions into PAs are predominantly located on islands, with higher alien richness in insular ($0.407 \pm 0.003$) than mainland ($0.252 \pm 0.003$) PAs overall, and within each taxon (Supplementary Fig. 3, Supplementary Table 2); the top three PAs in terms of alien richness are all on Hawaiian islands, in Volcanoes National Park (80 species), Hakalau Forest National Wildlife Refuge (63 species) and Kipuka Ainahou (62 species). Network analyses show that the geographic flows of alien animal species established in PAs are dominated by exchanges between areas within continents, and between South/East Asia and Europe (Fig. 3). Overall, South/East Asia and Africa are major donor regions of these aliens, and Europe and North America are the most important recipient regions (Fig. 3).

**Drivers of alien animal richness in PAs.** There is not a simple relationship between the number of established alien animal species in a PA and its IUCN conservation category, with the highest levels of incursion observed in supposedly relatively well protected IUCN level-II parks for most ecoregions across taxa (Fig. 4, Supplementary Fig. 4). By contrast, less stringent PAs have moderate richness of alien animals, with high levels of incursions only in certain biomes (e.g. mangroves) suffering intensive human pressures (Fig. 4, Supplementary Fig. 4). As most (~90%) PAs have no established alien animal species yet, we used mixed-effect models with a zero-inflated negative binomial distribution (ZINB) to test the significance of four abiotic, biotic and human activity factors in explaining their variation in alien

species richness. As there may be variation in the number of PAs and sampling effort among regions[24], we included country identity as a random factor to account for the spatial pseudo-replication[27]. Our results show that a PA's surface area, human footprint index, and year of designation are significant correlates of alien richness across taxa (Fig. 5): there tend to be more alien animals in PAs with larger areas, greater human footprint index and more recent designation as a PA. However, we do not obtain congruent results on the relationship between native biodiversity and established alien richness across taxa: there are negative associations between native and alien richness for reptiles and birds, but positive associations for mammals and amphibians (Fig. 5). We obtain similar results on the relative importance of predictors at different spatial resolutions (5 arc-min: Fig. 5; 2.5 arc-min and 10 arc-min: Supplementary Fig. 5), suggesting that the results were robust to potential uncertainties from this source.

## Discussion
Our present study, to the best of our knowledge, provides by far the most comprehensive global analysis of the relationship between PA designation and terrestrial alien faunas. We show that PAs have to date generally been effective in resisting alien animal incursions, with more than 90% of PAs still free of any of the 894 species in our database. This reinforces a recent temporal analysis that threats from animal invaders overall remained stable after 30 years in 21 case-study PAs[17]. One potential explanation is that most PAs are designated in areas with low human activities, which would have reduced opportunities for alien species to be introduced[28]. Alternatively, PA managers may have realised the impacts of alien species[29] and

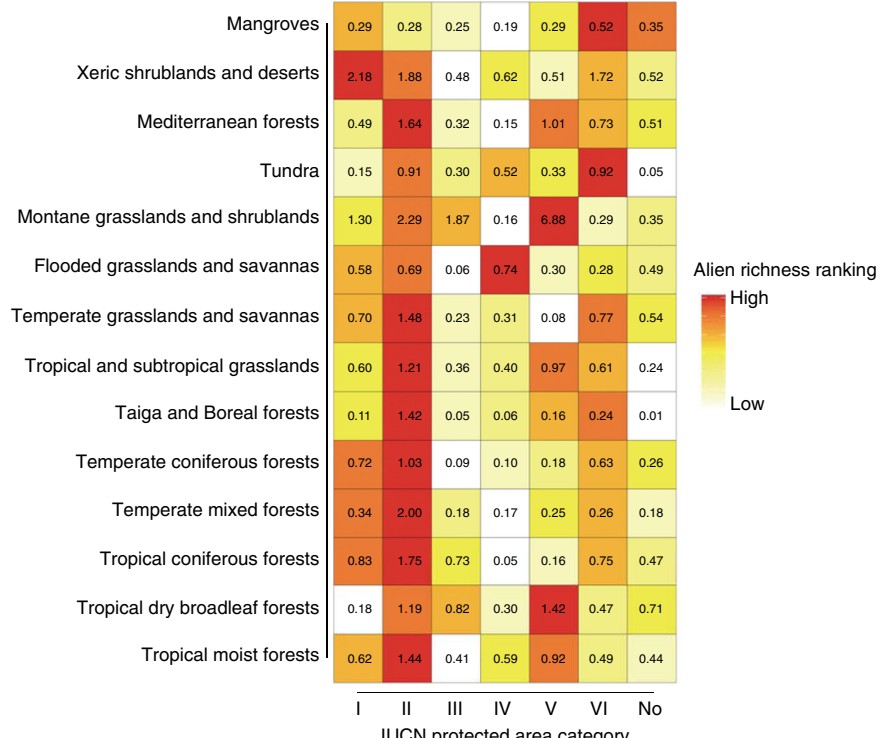

**Fig. 4 Heatmap showing the ranking of average richness of established alien animal species across IUCN conservation categories in 14 global biomes.** The colour scale was determined by ranking the average established alien animal species richness (in brackets) of PAs with different IUCN conservation categories in each of 14 biomes. Detailed values for each taxon are provided in Supplementary Fig. 4.

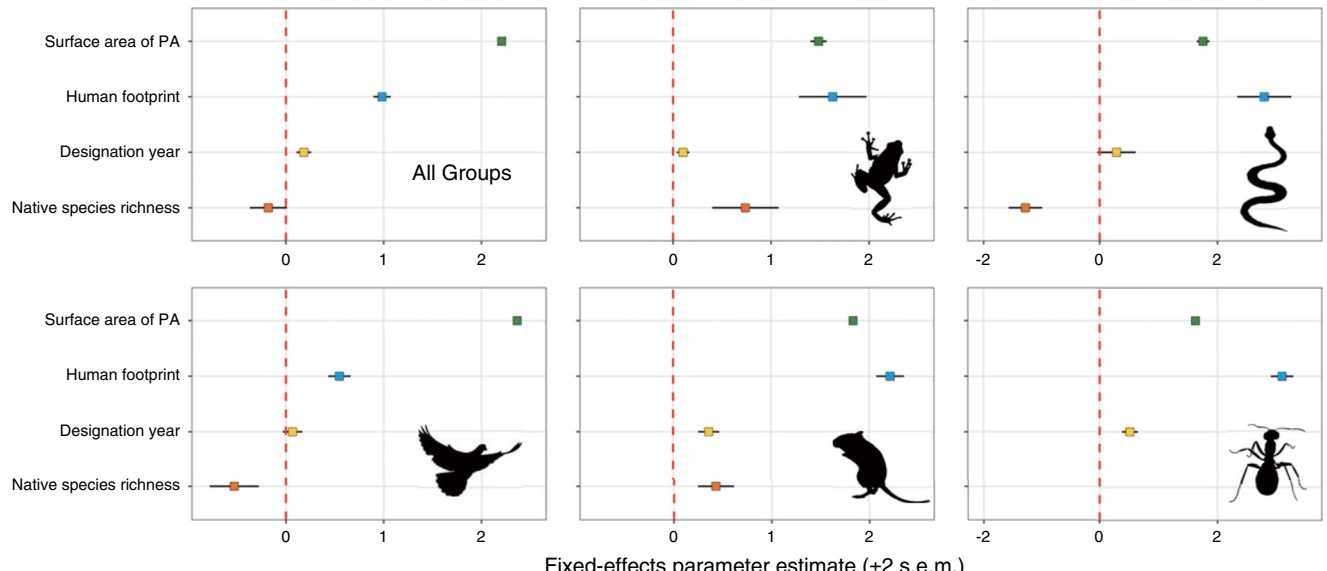

**Fig. 5 Fixed-effect parameter estimates of surface area, designation year, native species richness and human footprint on richness of established alien animal species across taxonomic groups.** Estimates of effects were obtained from zero-inflated negative binomial mixed-effects models with four abiotic, biotic and human activity factors as fixed effects and country identity as a nested random effect. Note that native species richness was used only in the model for vertebrate groups due to the unavailability of invertebrate native distribution data. A constant of 1 was added to each variable prior to log transformation. Credit: Tianjian Song (animal silhouettes).

strengthened strategies to prevent their introduction and pursue eradication[17]. However, more than 95% of global PAs are predicted to have high habitat suitability (Fig. 1b), while the presence of c.80% of the alien animals in our study within 10 km of their boundaries suggests that PAs may be vulnerable to the accelerating establishment and spread of alien species from surrounding areas, especially large PAs with high levels of

anthropogenic pressure. While the resolution at which the species distribution modelling was conducted here (10 arc-min, or ~20 km × 20 km at the Equator) is relatively coarse compared with higher resolution analyses at the continental scale[14], this resolution is justifiable for modelling global distributions, especially for birds with wide geographical ranges[30]. With the development of supercomputer and the increasing capacity of

high-resolution computation, it should be possible to make finer-scale predictions in the future.

Fewer alien animals have established in PAs designated longer ago (Fig. 4), which further validates the performance of PAs in restricting incursions[14,31]. This effect is particularly strong for herptiles, mammals and invertebrates with relatively low natural dispersal abilities, which are mainly transported by humans through the pet trade and as stowaways[32,33], and for which human footprint was the most important predictor of richness in PAs (Fig. 5). Year of designation is a relatively weak predictor of alien bird richness; for this taxon, PA area is the most important predictor (Fig. 5), possibly reflecting the difficulty of preventing natural dispersal by highly mobile volant species. Older PAs are more likely to be located in remote areas far from human disturbance[28], which may also explain these effects. Our study does not suggest that PAs consistently act a barrier to alien invaders owing to their rich resident communities[31], echoing the controversial relationship between native diversity and invasion outcomes in the literature[23]. However, there is a negative signal for alien reptiles and birds, the latter of which supports a recent study detecting a negative effect of native avian richness on the establishment success of alien birds worldwide[34]. PAs on islands, which tend to be more species-poor, are also more heavily affected by alien animals than mainland PAs, concurring with previous global syntheses not restricted to PAs[24], which may further explain why islands are foci of invasion-related species extinctions[35], especially for PAs in the Hawaii Islands, which have also been observed to harbour a high richness of alien plants[36].

The high habitat suitability within PAs, proximity to surrounding established populations, and the important role of human footprint index in predicting alien animal richness, imply that biological invasions are an impending threat to most PAs. Unfortunately, to our knowledge, comprehensive interventions against alien species incursions have not been considered in the traditional conservation planning of PAs, which is reflected by our finding that the current IUCN protection category for PAs does not fully reflect their establishment status. For example, one target for IUCN category-II National parks is to reduce biological invasions[15]. However, we show that the highest richness of alien animals is observed in such parks. National parks have relatively low human footprint values[5] (Supplementary Fig. 6), implying that other human activities, such as tourism, may drive alien richness in parks[13,37] but might not have been fully captured by this metric; this warrants future investigation once global tourist data in PAs are available. Alternatively, the surface area of national parks is much greater than other PAs (Supplementary Fig. 6) because they are always designed with large areas to maintain large-scale ecological processes[15], which may explain why national parks harbour more alien animal species. The conventional primary goal of PAs mainly involves protection of endangered species, their habitats and maintenance of ecological and evolutional process[15]. However, PAs have been regarded as important refuges for biodiversity under global environmental change[2,14,38], and so our results highlight the urgent importance and necessity of evaluating threats of these global change factors, such as alien species as presented here.

Moreover, our focus here on terrestrial alien animals is likely to underestimate the extent of alien species incursions in PAs worldwide: invasions are likely to be higher once information on alien aquatic animals and plants is incorporated, as such alien species are more prevalent than terrestrial alien animals[24]. For instance, it is notable that all case-study PAs in the SCOPE programme have been invaded by alien plants[16] and the threats from plant invaders have continued to increase over the past 30 years[17]. In addition, there might have been a geographical bias in sampling efforts in occurrences of alien species[39]. Most invasion

studies in PAs are conducted in Europe, the USA, South Africa and Australia, with many fewer in other regions, especially those undeveloped and developing countries. As additional data become available, we expect further studies across more taxonomic groups to support or refute our conclusions.

Although scientists and organisations have been trying to raise awareness of alien species problems in PAs for many years[16], our results demonstrate that more actions are needed to mitigate established alien species and to maintain PAs core functions in protecting native biodiversity. Routine monitoring of new introductions from visitors and vehicles entering parks is a key task to prevent the arrival of alien species. Increasing efforts to monitor, identify and eliminate early propagules in PAs and nearby areas are also needed and may be particularly important for those species with strong dispersal abilities such as birds. Remarkably, the high numbers of established alien populations in areas close to PAs emphasizes the need for prompt actions to control rising introduction and spread risks of these populations, especially for those PAs in proximity to high human populations, and that are connected with rivers and roads as important dispersal corridors[40,41]. There is thus an urgent need to integrate efforts from the scientific community, governments, NGOs, landowners and local stakeholders to develop more effective biosecurity strategies to pre-empt potential further invasions in PAs under ongoing global change.

## Methods

**Global protected area data**. We obtained data on the location, boundary, designation year and area of global protected areas from the April 2019 version of the World Database on Protected Areas (WDPA)[42], which provides unified standards and unique opportunities for large-scale conservation studies[5,43]. Although there is variation in number of PAs in the WDPA, we did not include more data from the country-level or regional database because the WDPA is the only authoritative dataset following globally consistent standards, and is regularly validated and updated to maintain the highest data qualities[44]. Following the standard of previous global studies[5,45], we only included PAs with detailed geographical information and a status of "designated" (not including those designated as UNESCO Man and Biosphere Reserves), "inscribed", or "established" for data analyses. As many PAs in the WDPA database have spatially overlapping polygons, we followed WDPA best practice guidelines (https://www.protectedplanet.net/c/calculating-protected-area-coverage) and dissolved the overlapped areas into a single polygon, assigning overlapping areas as the stricter IUCN conservation category[5,43]. In addition, we applied the simplify function in ESRI ArcGIS 10.2.1 to remove redundant vertices, with a tolerance of 1 km to facilitate computation[5]. As previous studies have suggested that global invasion hotspots are mainly located on islands[19,24,46], we compared the establishment patterns of alien animals between insular and mainland PAs by categorizing them as being completely located on islands or continental mainland following a new 30-metre resolution global shoreline vector and associated islands database[47], downloaded from USGS Global Ecosystems website (https://rmgsc.cr.usgs.gov/ecosystems/datadownload.shtml).

Global protected areas have different International Union for Conservation of Nature (IUCN) categories, ranging from I for strictly protected areas to VI for protected areas with sustainable use of natural resources, and those termed not reported, not applicable and not assigned[15]. To evaluate whether the present IUCN categories reflect alien animal incursion states, we compared alien species richness among different categories. Variation in habitat types among categories may further influence the spatial distributions of alien species. We therefore followed previous studies[5] by extracting 14 global biomes derived from spatial distributions of 867 terrestrial ecoregions across the globe[48], and then compared the alien species richness among IUCN categories in each biome.

**Alien species and occurrence data**. We analysed a total of 894 terrestrial alien animal species from 11 taxa including four vertebrate taxa (Amphibia, Reptilia, Aves and Mammalia) and seven invertebrate taxa (Arachnida, Branchiopoda, Chilopoda, Diplopoda, Malacostraca, Gastropoda and Insecta). We identified the establishment status of each alien species and collected their occurrence data across the globe from different databases and references (Supplementary Data 1-2). For terrestrial vertebrate taxa, we determined the study species and their exact native and invaded ranges based on the IUCN (www.IUCN.org, accessed on January 12, 2018), Kraus' compendium[32] and others following updates for amphibians and reptiles[49,50], the BirdLife International & NatureServe geodatabase (BLINS, available at http://datazone.birdlife.org/species/requestdis, accessed on January 12, 2018) and the Global Avian Invasions Atlas (GAVIA) for bird invasions[51], Long's book (2009)[33] and the update by Capellini et al. for the alien mammal species[52].

These databases and references have been widely used in previous global studies describing the presence, origin, establishment and spatial distributions of different taxa around the world. Compared with vertebrates, most alien invertebrates generally lack exact information of native ranges[53]. We thus used 77 terrestrial alien invertebrates across 7 taxa with definite native and invaded ranges from the Global Invasive Species Database (GISD, http://www.iucngisd.org/gisd/, accessed on May–July, 2019). For all species, we carefully checked their geographic and taxonomic accuracy and excluded those species without precise distribution data, or those non-breeding populations such as migratory birds. We finally used a total of 98 amphibians, 178 reptiles, 391 birds, 150 mammals and 77 invertebrates for further analyses. To calculate the richness of alien animals in each PA, we used the occurrence data with precise coordinates indicating their presence in the PA for each species except for birds. For birds, as it is very difficult to obtain occurrences due to their strong flying abilities, we followed previous global studies calculating alien avian richness by using the GAVIA distribution maps[51] representing the minimum ranges of breeding populations of alien birds and a species were regarded present in a PA if any of its established alien range fell in the PA boundary[24,54]. Additionally, we observed both a high correlation (Pearson coefficient $r = 0.95$) and a high consistency (linear regression $R = 0.94$) in richness of alien birds within PAs between alien bird presence data derived from GAVIA maps and occurrence coordinates collected from databases and supplementary references (Supplementary Data 1 and 2), demonstrating that data sources used to calculate bird richness might not influence our results. Finally, there was no significant difference (Wilcoxon rank test for 17 shared case-study PAs, $Z = -0.298$, $p = 0.766$) in alien animal richness between our study and a recent global study based on a questionnaire survey to PA managers[17], which further validated the robustness of our alien richness estimation in each of PAs. The total alien species richness was calculated by summing all species across taxa established within each PA and was mapped in ArcGIS. To evaluate the potential spread risks of alien animals from surrounding areas to the PAs, we additionally identified whether there were established populations of each alien animal around each PA within a distance of 10 and 100 km to the PAs' boundaries using the Near function in ArcGIS. Based on the native and alien range information for each species, we conducted network analyses to quantify the global flows of those alien animals successfully established in PAs and identify their major donor and recipient regions using the Circlize package in R version 3.6.2 (R Development Core Team, 2019). (Supplementary Method 1), following the procedures of previous studies[55].

**Explanatory variables of established alien animals in PAs**. We explored the relationship between the richness of alien animal species established in each PA with four major factors encompassing abiotic conditions, biotic variables and human activities, which may potentially influence alien animal establishment. These included the year of designation of PAs from the WDPA database and their dissolved surface area calculated using ArcGIS. In addition, we extracted the native species richness for amphibians, mammals and birds from the Biodiversity Mapping website (https://biodiversitymapping.org/wordpress/index.php/home/). These maps are based on data from the IUCN for mammals and amphibians and BirdLife International for birds, and plot global grid-based richness for each taxon at a spatial resolution of 10 km × 10 km[56]. For reptiles, we used the recent updated database of global reptile spatial distributions[57] and generated the grid-based richness using Spatial Join functions in ArcGIS. As global maps on invertebrates are not yet available for most taxa, we did not include this variable for alien invertebrate analysis. We used human footprint index as a proxy to reflect the anthropogenic processes that may facilitate alien species invasions, such as roads, railways, waterways and human population density at a 1 km² spatial resolution from the Last of the Wild Project website[58], and then were adjusted to the native richness resolution (i.e., 5 arc-min, or ~10 km × 10 km). The value of the human footprint index ranges from 0 to 100 according to the degree of human influence[58], and we calculated the mean human footprint for each PA based on values of grid cells overlapping with PA boundaries[5]. As the spatial resolution at which the analyses were conducted might affect the results[59], we additionally performed analyses at both a finer resolution (2.5 arc-min, or ~5 km × 5 km at the Equator) and a coarser resolution (10 arc-min, or ~20 km × 20 km at the Equator) to evaluate the generality of our results at different spatial resolutions.

Because large numbers of PAs have not been invaded by alien animal species, we used zero-inflated negative binomial distribution regression (ZINB) to test the significance of the predictor variables in explaining the richness of established alien animals in PAs[14]. ZINB is especially useful for our database, which has an excess of zero counts that always have overdispersion issues, by assuming that the response variable is a function of a binomial process (established or not) and a count process (negative binomial distributed richness). Furthermore, as there is considerable regional variation in the number of PAs, with more than half of global terrestrial PAs located in Europe (55.8%) and North America (20.6%), and because sampling effort for alien species may vary greatly among regions[24], we treated region identity as a random effect to account for possible issues of spatial pseudo-replication[27]. We used the glmmTMB function in the glmmTMB package in R[60] (Supplementary Method 2).

**Habitat suitability of PAs hosting alien animals**. To quantify the potential habitat suitability of global PAs for alien animal establishment, we applied species

distribution modelling (SDM), which is a commonly used and powerful tool to quantify habitat suitability for alien species[25,61]. We used *Biomod2* to predict suitable habitats for each of the 894-alien species using an ensemble of five widely used SDM algorithms including boosted regression trees, generalized additive models, multiple adaptive regression splines, classification tree analysis and random forest[62] (Supplementary Method 3). SDMs fit statistical relationships between the species' geographic distributions and the corresponding habitat predictors, with a higher habitat suitability value for a given grid cell indicating a higher relative probability of species' presence[63].

We developed SDMs using occurrence data from both of species native and invaded ranges (Supplementary Data 1 and 2) in order to avoid underestimating a species' entire niches, because alien terrestrial animals can occupy novel realised niches in new ranges[64–66]. Although climate is widely used as one fundamental predictor of alien species establishment[34], we also incorporated microhabitat factors including vegetation and water availability that may directly and indirectly influence species distributions by affecting food availability, reproduction and biotic interactions, which are increasingly realised as important to SDM performance[67]. We used different bioclimatic variables known to be physiological constraints for different taxa from the WorldClim-Global Climate Database (www.worldclim.org)[68]. For amphibians and reptiles, we used a total of eight temperature and precipitation variables: annual average temperature and precipitation, seasonal temperature and precipitation, the minimum temperate of the coldest month, the highest temperature of the warmest month, and the precipitation of the wettest and the driest quarters[69]. For birds, we used six bioclimatic variables: temperature seasonality, maximum temperature of warmest month, minimum temperature of coldest month, precipitation of wettest month, precipitation of the driest month and precipitation seasonality[62,70,71]. For mammals, we used a total of 10 bioclimatic variables based on previous studies of mammal species distribution modelling at large spatial scales:[72,73] annual mean temperature, mean temperature of the wettest quarter, mean temperature of the driest quarter, mean temperature of the warmest quarter, mean temperature of the coldest quarter, annual precipitation, precipitation of the wettest quarter, precipitation of the driest quarter, precipitation of the warmest quarter, precipitation of coldest quarter. For invertebrates, we used mean diurnal range, max temperature of warmest month, min temperature of coldest month, annual precipitation, precipitation of wettest month, and precipitation of driest month[61,64]. We used the annual normalised difference vegetation index (NDVI) based on the monthly data from years of 2001–2005 (http://neo.sci.gsfc.nasa.gov/) for the vegetation variable[74]. For water availability, we extracted the open waters from the global lakes and wetlands database (GLWD, https://www.worldwildlife.org/pages/global-lakes-and-wetlands-database) by combining layer GLWD-1 (the largest lakes and reservoirs) and layer GLWD-2 (other lakes, reservoirs and rivers with a surface area ≥0.1 km²) and removed saltwater lakes based on the information from the Saline Lakes database (http://lakes.chebucto.org/saline1.html)[26]. All predictor variables we used lack significant collinearity problems, as coefficients of these variables were all <0.75 across taxon according to pairwise Spearman rank correlation analyses. All species occurrence data and environmental variables were projected onto a Behrmann equal-area cylindrical projection (~10 × 10 arc-min at the equator) to account for the potential effect of grid cell area changes with latitude[75]. This spatial resolution covers 90% of the total area of global PAs and can process within the computer calculation ability for global large dataset analyses.

The sampling effort in occurrence data of alien species may vary greatly across taxa among PAs, which may influence the prediction results of SDMs[76]. We applied a target-group method to minimise potential sampling bias on our results[77]. All occurrence data from the Global Biodiversity Information facility (GBIF, http://www.gbif.org) for each taxon were used as the background data representing available sampling areas to account for the distribution of sampling effort for each taxon across the globe[77]. Target-group method allows background data having the same potential bias as the occurrence data, and has been shown a good performance to deal with sampling bias issue in SDMs[78]. Considering the different sample sizes among taxa, ranging from relatively small range sizes for amphibians and reptiles to wider distributional ranges for mammals, birds and invertebrates, we randomly chose 30,000 background data points for amphibians and reptiles, 70,000 background data for mammals, 100,000 for birds, 80,000 for invertebrates to run each SDM[79]. Equal weights were given to presence data and background points (i.e., 50% balancing the weights of presences and background points to a prevalence of 0.5)[25,79]. We followed previous studies by using a threshold maximising TSS method to convert continuous SDM outputs into species presence (1) and absence (0) predictions, and then estimated the total number of species for each grid cell by summing the resultant presence-absence maps[62]. We determined the number of potential invaders of PAs based on the predicted richness of alien species in each grid with which the PAs are intersected. We applied an ensemble approach to reduce prediction variation by different SDM algorithms[80]. In order to increase model prediction accuracy, we excluded those models with AUC < 0.8 or TSS < 0.6 from the final ensemble prediction[25]. We assigned weights to each model based on their TSS values and constructed ensemble models by calculating the weighted mean of environmental suitability across the predictions[25]. When SDMs are projected to new geographic regions, there are usually non-analogous climates (i.e., areas where the value of at least one predictor variable is outside the training region)[81]. In order to minimise such uncertainties, we made conservative predictions and restricted our model

projections onto those analogous climates that can be sampled by occurrence and background records in both native and invaded ranges.

**Reporting summary**. Further information on research design is available in the Nature Research Reporting Summary linked to this article.

## Data availability

The study species list is provided in Supplementary Data 1 and 2. The list is based on the IUCN (www.IUCN.org), Kraus' compendium[32] (https://link.springer.com/chapter/10.1007/978-1-4020-8946-6_6), Capinha et al.[49] (https://doi.org/10.1111/ddi.12617), Liu et al.[50] (https://doi.org/10.1016/j.cub.2018.12.036), the BirdLife International & NatureServe geodatabase (http://datazone.birdlife.org/species/requestdis), the Global Avian Invasions Atlas (https://doi.org/10.6084/m9.figshare.4234850), Long's (2009) book (https://ebooks.publish.csiro.au/content/introduced-mammals-world), Capellini et al.[52] (https://doi.org/10.1111/ele.12493) and Global Invasive Species Database (http://www.iucngisd.org/gisd/). All occurrence data used in this paper are freely available from the web databases including GBIF (http://www.gbif.org/), ALA (http://www.ala.org.au/), Arctos (http://arctos.database.museum/), CBIF (http://www.cbif.gc.ca/), EUNIS (http://eunis.eea.europa.eu/), HerpNET (the original HerpNET2 website has been migrated to the VertNet portal http://www.vertnet.org/search on January 5, 2015), iNaturalist (http://www.inaturalist.org/), NMNH (https://naturalhistory.si.edu/research/vertebrate-zoology), NCMNS (http://collections.naturalsciences.org), SpeciesLink (http://www.splink.org.br/), USGS Nonindigenous Aquatic Species Database (http://nas.er.usgs.gov/), Vertnet (http://portal.vertnet.org/), ORNIS (http://ornis2.ornisnet.org/), Global Avian Invasions Atlas (https://doi.org/10.6084/m9.figshare.4234850), BirdLife International & NatureServe (http://datazone.birdlife.org/species/requestdis), and an intensive review of 959 published references (Supplementary Data 1 and 2). We obtained data on the location, boundary, designation year and area of global protected areas from the World Database on Protected Areas (www.protectedplanet.net). Human footprint data are available from the Last of the Wild Project website (https://sedac.ciesin.columbia.edu/). Native amphibian, bird and mammal richness data are available on from Biodiversity Mapping website (https://biodiversitymapping.org/wordpress/index.php/home/). For reptile, native distributional range maps are extracted from Dryad (https://doi.org/10.5061/dryad.83s7k). All climatic data are available on the WorldClim-Global Climate Database (www.worldclim.org). Vegetation data are available on the Nasa Earth Observations (https://neo.sci.gsfc.nasa.gov/). Water data are available on the global lakes and wetlands database (GLWD, https://www.worldwildlife.org/pages/global-lakes-and-wetlands-database).

## Code availability

The R code for running the entire analysis is available in the Supplementary Information (Supplementary Method 1–3).

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

## Acknowledgements

Y.L. and X.L. were supported by grants from Second Tibetan Plateau Scientific Expedition and Research (STEP) Programme (2019QZKK0501), National Science Foundation of China (31530088 and 31870507) and Youth Innovation Promotion Association of Chinese Academy of Sciences (Y201920).

## Author contributions

Y.L. and X.L. designed the study; X.L., X.W., T.S., C.H. and Y.L. collected the data; X.L., T.S. and Y.L. analyzed the data; X.L., T.M.B. and Y.L. wrote the manuscript.

## Competing interests

The authors declare no competing interests.
