## [Peer Review File · Nature Communications]

Reviewers' comments first round:

Reviewer #1 (Remarks to the Author):

LIU ET AL NATURE COMMUNICATIONS MS: "ANIMAL INVADERS THREATEN PROTECTED AREAS WORLDWIDE"

This paper addresses a topical issue. Animal invasions in protected areas are indeed a major and growing problem – an important paper that could be cited to justify this study is that of Shackleton et al. (2020). It would also be useful to compare the results of this study with those of the Shackleton study which reports on changes in the presence of alien species in protected areas.

65 I have studied the methods applied in the paper and found them to be appropriate. It was good to see that different variables were used to compute habitat suitability of different major taxonomic groups. The description of the limitations of the data and the resulting caveats looked good to me.

The presentation of results looks good. Figure 3, in particular, is useful (perhaps explain what high/low alien richness ranking means though?)

Reference cited in the review:

Shackleton, R.T. et al. (2020). Assessing biological invasions in protected areas after 30 years: Revisiting nature reserves targeted by the 1980s SCOPE programme. *Biological Conservation* (in press) <https://doi.org/10.1016/j.biocon.2020.108424>

Reviewer #2 (Remarks to the Author):

This ms is definitively a powerful paper grounded on a massive analysis of open data worldwide.

While I appreciate the workflow effort, I should also report a basic lack of a proper Discussion of the achieved results, in view of previous analysis done by other researchers. Hence my first suggestion is to reinforce that part.

Moreover, I would definitively add the code being used to perform the whole analysis, from data gathering until the spatial output. *Nature Communications* is a very good sounding board for spreading the idea about the potential invasion of animal species. Hence, providing the whole analysis flow (including the code) would be useful for several researchers.

Besides, no spatial information is provided about the uncertainty of the achieved results. It is well known that most of the datasets used in this manuscript and in several other examples are biased from several points of view, as any worldwide set should actually be: e.g. different sampling effort or problems in taxonomic determination which cannot be controlled... Hence, I would expect a spatial estimate of the final uncertainty of the achieved results.

Minor points:

Suppl. Figure 3 is impressive and well done. It definitively deserves to be part of the main manuscript.

The following link is broken:

<https://www.protectedplanet.net/c/calculating-protected-areacoverage>

It does not work even when removing '/c/'.

Best regards.
Duccio Rocchini

--

Duccio Rocchini, PhD

Professore Ordinario | Full Professor
Alma Mater Studiorum University of Bologna
Dept. Biological, Geological and Environmental Sciences
<https://www.unibo.it/sitoweb/duccio.rocchini/>

Reviewer #3 (Remarks to the Author):

This manuscript presents the most comprehensive assessment today about animal invasions in protected areas worldwide. Considered one of the 5 most important drivers of biodiversity loss globally (IPBES Global Assessment 2019), the threat of biological invasions is severely underestimated in protected areas. This study provides compelling evidence that 89.4% of protected areas worldwide have established invasive populations in their buffer areas (<10 km). Even if only 10% are considered officially invaded (likely a huge underestimation of the true figure), the study demonstrate that invasions may become an increasingly dominant problem in the near future. The study links animal invasions in protected areas to the date of designation, area, and human influence, and summarize levels of invasion by IUCN category, taxonomic group and habitat. This information is extremely important to guide future research as well as the design of prevention and early action plans. The amount of data and analyses behind this manuscript is impressive, and the result is not only novel but also useful, since authors make a number of management recommendations arising for this research. The manuscript is very well written, easy to follow, and methods are rigorous and well explained.

Line 48. Actually, one of the main conclusions of our study (Gallardo et al., Protected areas offer refuge from invasive species spreading under climate change, 2017) is that Natura 2000 protected areas, which are more recently designated and less strictly protected, harbored significantly more invasive species than national parks at the continental scale.

Lie 329. Can you describe how did you generated the vegetation and water availability maps? I couldn't find this information.

Line 353. 10x10 minute resolution. My one and only concern with this manuscript is the low resolution chosen for some of the analysis that approximates (if I am not mistaken), 20x20 km at the equator. I understand the trade-off between resolution and scale, overall considering the high number of species analyzed, but I am concerned it may be too gross to characterize the conditions of PPAA, let alone their water availability or vegetation. Considering the increasing computational capacity of most research institutions, I'd strongly advice aiming for a higher resolution in future studies.

Congratulations, I am looking forward to see this study published.
Dra Belinda Gallardo

Point by Point Response to Reviewer Comments:

Reviewer #1 (Line numbers refer to revised MS and all changes have been highlighted with blue color)

Point 1) *This paper addresses a topical issue. Animal invasions in protected areas are indeed a major and growing problem – an important paper that could be cited to justify this study is that of Shackleton et al. (2020). It would also be useful to compare the results of this study with those of the Shackleton study which reports on changes in the presence of alien species in protected areas.*

Our Response: We thank the reviewer for their positive comments emphasizing the importance of our present work, and we very much appreciate them providing the latest publication on this topical issue. We have cited this recent paper in our revised manuscript (Line 73-75, Line 85, Line 180-182, Line 184-186). Importantly, we observed a high coincidence in the alien vertebrate richness between the shared PAs used in Shackleton et al. (2020) and our present study, validating the robustness of our quantification on alien richness in PAs worldwide (Line 337-341).

Point 2) *I have studied the methods applied in the paper and found them to be appropriate. It was good to see that different variables were used to compute habitat suitability of different major taxonomic groups. The description of the limitations of the data and the resulting caveats looked good to me.*

Our Response: Thank you very much for your careful reviews of our method, and we are pleased to have your approval.

Point 3) *The presentation of results looks good. Figure 3, in particular, is useful (perhaps explain what high/low alien richness ranking means though?)*

Our Response: We apologize that we did not describe the legend of Fig. 3 (now Fig. 4) clearly. The high-low ranking is the order of seven IUCN conservation categories based on their average alien richness in each of 14 biomes. In our revised version, we have provided the exact value of average alien richness in each category across 14 biomes (Line 751).

Reviewer #2

Point 1) *This ms is definitively a powerful paper grounded on a massive analysis of open data worldwide. While I appreciate the workflow effort, I should also report a basic lack of a proper Discussion of the achieved results, in view of previous analysis done by other researchers. Hence my first suggestion is to reinforce that part.*

Our Response: Thank you very much for your positive comments on our work, and for the helpful suggestions to reinforce the Discussion. Following your suggestions, we have further strengthened our manuscript by discussing more about how our present study relates to previous studies (Line 180-182, Line 184-186, Line 216-217, Line 242-245, Line 258-262).

Point 2) *Moreover, I would definitively add the code being used to perform the whole analysis, from data gathering until the spatial output. Nature Communications is a very good sounding board for spreading the idea about the potential invasion of animal species. Hence, providing the whole analysis flow (including the code) would be useful for several researchers.*

Our Response: Thank you very much for this very good suggestion. We have provided all analysis flow and code in data analyses as one supplementary figure (Supplementary Fig. 1) and one supplementary table (Supplementary Table 3) for the analysis flow, and three supplemental tables for code and model formula (Supplementary Tables 5-7).

Point 3) *Besides, no spatial information is provided about the uncertainty of the achieved results. it is well known that most of the datasets used in this manuscript and in several other examples are biased from several points f view, as any worldwide set should actually be: e.g. different sampling effort or problems in taxonomic determination which cannot be controlled... Hence, I would expect a spatial estimate of the final uncertainty of the achieved results.*

Our Response: The reviewer makes a very constructive point here relating to the potential uncertainties of our work. In order to estimate the uncertainty due to spatial resolution, we re-conducted our analyses at another two spatial resolutions (a finer 2.5 arc-minutes and a coarser 10 arc-minutes scale where the SDM analyses were conducted, respectively) and obtained similar results with our main analyses at a resolution of 5 arc-minutes, indicating that our results were robust to spatial

uncertainties. We have clarified the spatial resolution issue in our revised manuscript (Line 369-371, Line 374-378), and have added one supplementary figure (Supplementary Fig. 5) showing the results of these new analyses. We completely agree with the reviewer that these gaps in knowledge are very difficult to control in large-scale spatial analysis. However, if we assume that these gaps are, for the most part, random in their distribution, they should not dramatically alter our conclusions, and we sincerely believe that the imperfect data at a global scale could not preclude us from attempting to answer questions at this scale.

Point 4) Minor points:

Suppl. Figure 3 is impressive and well done. It definitively deserves to be part of the main manuscript.

Our Response: Thank you. We have moved Supplementary Fig. 3 to the main text (new Fig. 3) accordingly.

Point 5) The following link is broken:

<https://www.protectedplanet.net/c/calculating-protected-areacoverage>

It does not work even when removing '/c/'.

Our Response: Sorry for this inconvenience. There was a “-” missing behind the ‘area’. We have corrected this typo error in our revised manuscript (Line 281).

Reviewer #3

Point 1) *This manuscript presents the most comprehensive assessment today about animal invasions in protected areas worldwide. Considered one of the 5 most important drivers of biodiversity loss globally (IPBES Global Assessment 2019), the threat of biological invasions is severely underestimated in protected areas. This study provides compelling evidence that 89.4% of protected areas worldwide have established invasive populations in their buffer areas (<10 km). Even if only 10% are considered officially invaded (likely a huge underestimation of the true figure), the study demonstrate that invasions may become an increasingly dominant problem in the near future. The study links animal invasions in protected areas to the date of designation, area, and human influence, and summarize levels of invasion by IUCN category, taxonomic group and habitat. This information is extremely important to guide future research as well as the design of prevention and early action plans. The amount of data*

and analyses behind this manuscript is impressive, and the result is not only novel but also useful, since authors make a number of management recommendations arising for this research. The manuscript is very well written, easy to follow, and methods are rigorous and well explained.

Our Response: Thank you very much for your positive comments and interest in our manuscript. We have also added the IPBES global report in our Introduction section as one important background knowledge of alien species impacts worldwide (Line 39-42).

Point 2) *Line 48. Actually, one of the main conclusions of our study (Gallardo et al., Protected areas offer refuge from invasive species spreading under climate change, 2017) is that Natura 2000 protected areas, which are more recently designated and less strictly protected, harbored significantly more invasive species than national parks at the continental scale.*

Our Response: Thank you. We have rephrased this sentence to make the original finding of the previous study more clearly and concise (Line 53-55).

Point 3) *Lie 329. Can you describe how did you generated the vegetation and water availability maps? I couldn't find this information.*

Our Response: We apologize for the missing of this important information. In our

revised manuscript, we have provided the source of vegetation and water variables in the Method section (Line 427-433).

Point 4) *Line 353. 10x10 minute resolution. My one and only concern with this manuscript is the low resolution chosen for some of the analysis that approximates (if I am not mistaken), 20x20 km at the equator. I understand the trade-off between resolution and scale, overall considering the high number of species analyzed, but I am concerned it may be too gross to characterize the conditions of PPAA, let alone their water availability or vegetation. Considering the increasing computational capacity of most research institutions, I'd strongly advice aiming for a higher resolution in future studies.*

Our Response: We completely agree with the reviewer that the resolution of species distribution modeling is coarser than the previous study at the continental scale. Following the reviewer's suggestion, we have added an acknowledgement in the Discussion section to address this limitation, and discuss this as a potential avenue for future research with increasing computational capacity (Line 191-197).